# Refining clinical algorithms for a neonatal digital platform for low-income countries: a modified Delphi technique

Mari Evans ![ORCID],[1] Mark H Corden ![ORCID],[2,3] Caroline Crehan ![ORCID],[1] Felicity Fitzgerald ![ORCID],[1] Michelle Heys ![ORCID] [1]

► Prepublication history and additional online supplemental material for this paper are available online. To view these files, please visit the journal online (http://dx.doi.org/10.1136/bmjopen-2020-042124).

ME and MHC are joint first authors.

[1]UCL Institute of Child Health, Great Ormond Street Hospital for Children, London, UK
[2]Division of Hospital Medicine, Department of Pediatrics, Children's Hospital Los Angeles, Los Angeles, California, USA
[3]Department of Pediatrics, University of Southern California Keck School of Medicine, Los Angeles, California, USA

**Correspondence to**
Dr Michelle Heys;
m.heys@ucl.ac.uk

## ABSTRACT

**Objectives** To determine whether a panel of neonatal experts could address evidence gaps in local and international neonatal guidelines by reaching a consensus on four clinical decision algorithms for a neonatal digital platform (NeoTree).

**Design** Two-round, modified Delphi technique.

**Setting and participants** Participants were neonatal experts from high-income and low-income countries (LICs).

**Methods** This was a consensus-generating study. In round 1, experts rated items for four clinical algorithms (neonatal sepsis, hypoxic ischaemic encephalopathy, respiratory distress of the newborn, hypothermia) and justified their responses. Items meeting consensus for inclusion (≥80% agreement) were incorporated into the algorithms. Items not meeting consensus were either excluded, included following revisions or included if they contained core elements of evidence-based guidelines. In round 2, experts rated items from round 1 that did not reach consensus.

**Results** Fourteen experts participated in round 1, 10 in round 2. Nine were from high-income countries, five from LICs. Experts included physicians and nurse practitioners with an average neonatal experience of 20 years, 12 in LICs. After two rounds, a consensus was reached on 43 of 84 items (52%). Per experts' recommendations, items in line with local and WHO guidelines yet not meeting consensus were still included to encourage consistency for front-line healthcare workers. As a result, the final algorithms included 53 items (62%).

**Conclusion** Four algorithms in a neonatal digital platform were reviewed and refined by consensus expert opinion. Revisions to NeoTree will be made in response to these findings. Next steps include clinical validation of the algorithms.

## INTRODUCTION

Globally, 2.5 million newborns die each year in the first 28 days of life.[1] Most of these deaths (98.5%) occur in low-income countries (LICs), and 40% occur on the first day of life.[2] The neonatal mortality rate has halved since 1990,[3] but modelling of global newborn mortality data suggests that a further two-thirds of current deaths could be prevented if evidence-based solutions were implemented.[2] One of the WHO Sustainable Development Goals is to end preventable deaths of newborns in all countries and to reduce the neonatal mortality rate from the current rate of 18 per 1000 live births to less than 12 per 1000 by 2030.[4] Targeting newborn care in LICs is thus an urgent priority, especially the three most common causes of mortality—infections (36%), prematurity (28%) and intrapartum complications (23%).[2]

The WHO neonatal guidelines are internationally recognised as the leading and most respected source of guidance.[5] However, one of their limitations is that they are primarily based on data from high-income countries, as there is often a lack of evidence in LICs due to limited diagnostic aids, data and research.[6] WHO aims to address the challenge of developing setting-appropriate neonatal guidelines by improving stakeholder involvement

**Strengths and limitations of this study**

► In this study, a large number of algorithm items were reviewed and evaluated, and half met consensus for inclusion in the management pathways.

► The review was conducted with experts from a broad range of countries and neonatal experience who simultaneously refined the algorithms and highlighted gaps in current evidence, emphasising the need for future research to support international neonatal guidelines.

► Our study method meant that experts were not able to meet in person, which might have promoted dialogue that would have allowed greater clarity in their collective opinion.

► The representation of neonatal experts from low-income countries was not as robust as from high-income countries, which may have led to an uneven evaluation of the algorithms.

(design guidelines for specific audiences), clarity of presentation (often guidelines are too long and technical) and attention to dissemination.[7] Mobile health (m-health) technology and digital platforms are potential approaches to implementing these measures and improving the quality of newborn care.[8]

An international team of researchers, clinicians and software developers in the UK, USA, Malawi, Bangladesh and Zimbabwe codesigned and codeveloped with Malawian and Zimbabwean heathcare workers (HCWs) a neonatal digital platform (NeoTree) for facility-based newborn care in LICs. It combines immediate digital data capture (which is shared with HCWs via local dashboards), evidence-based algorithmic clinical decision and management support, newborn education and data linkage to national data systems on one platform.[9] The algorithms in the Malawian version of the NeoTree support decisions according to established international[10] and Malawian neonatal guidelines.[11] In situations where guidelines were not applicable, the NeoTree clinical team used clinical judgement to complete the algorithm development. In the absence of extensive trial or epidemiological data in LICs, alternative techniques to consolidate best available low-quality evidence can be used, such as expert opinion. This study aims to use the modified Delphi technique to determine whether a panel of experts in newborn care can reach a consensus opinion about key clinical decision algorithms used in a digital platform to assist HCWs caring for facility-based unwell newborns in LICs.

## METHODS
### Study design
This study used a two-step modified Delphi technique.[12] The Delphi technique was chosen because it is an effective method of gathering expert knowledge from geographically diverse leaders in the field to address complex clinical problems that lack evidence.

### Recruitment
Twenty-two neonatal experts were invited to participate in the study. This number represented an adequate sample size[13 14] and permitted a manageable amount of data collection. Participants were recruited if they were a physician or neonatal nurse practitioner with more than 10 years neonatal experience (at least three in LICs), neonatal postgraduate training, fluency in English, internet access and willingness to participate. Neonatal experts known to the researchers for their clinical expertise, research and contributions to guideline development in LICs were identified in equal numbers from both LICs and HICs. No financial incentive was offered, but reimbursement for costs of Skype calls was provided for some experts in LICs.

### Algorithms and item generation
The four clinical decision algorithms selected for review were neonatal sepsis, hypoxic ischaemic encephalopathy (HIE), respiratory distress of the newborn and hypothermia. These conditions represent the leading preventable causes of neonatal mortality and are the most difficult to diagnose and manage appropriately in LICs with some of the weakest WHO grade recommendations and quality of evidence.[15] For example, the European definition of neonatal sepsis is two or more clinical symptoms and two or more laboratory signs in the presence of, or as a result of, suspected or proven infection.[16] This definition is not possible in LICs where laboratory investigations are not routinely available.[17]

Items were identified by comparing the algorithms side by side with the international (WHO) and local neonatal guidelines (Care of the Infant and Newborn in Malawi—COIN) from which they had been derived. This comparison generated a comprehensive list of items where discrepancies in diagnostic parameters and treatment recommendations required expert opinion. Once finalised, the clinical algorithms and list of items (henceforth referred to as questionnaire) were piloted with two paediatricians with neonatal experience in LICs. Ambiguous items were amended accordingly.

### Delphi technique
The questionnaire was circulated by email to the experts with specific instructions at least 2 weeks before they were interviewed. Each algorithm was verbally and diagrammatically explained with their references specified (ie, WHO, COIN or NeoTree research team) to aid in decision-making during the interview (online supplemental file 1). Round 1 interviews were conducted in June and July 2018. Experts were sent up to two reminder emails to schedule their phone or Skype interview. Interviews were conducted privately from a home office. Standardised questions were used to review each item from the questionnaire. Experts were asked to rate their level of agreement for including an item in an algorithm using a five-point Likert scale. A five-point scale was chosen because evidence suggests that a five-point scale appears to be less confusing than a seven-point scale and to increase expert response rate.[18] Each rating was followed by open-ended questions to obtain the experts' rationale for their response and any amendment or additional items they would propose. All interview data were transcribed using both audiorecordings and notes made during the interview by the facilitator. All responses were anonymised (with participant numbers) and reviewed together with the quantitative results.

The upper limit of agreement among experts has been recommended to be set at 80% (4 or higher on the Likert scale) for Delphi studies.[13] Due to our sample size, this upper limit was used to apply greater rigour to item inclusion. Items that met consensus (≥80% agreement) were included or were modified with minor changes to wording based on expert advice. Items that did not meet consensus (<80%) were removed or modified according to the feedback from the expert panel and submitted for the second round. Items that did not meet consensus

were still included if they were part of WHO and COIN guidelines so that frontline HCWs continued to follow the current standard of care. A second questionnaire was designed with modified items and expert additions from the first round (eg, inclusion of the Thompson encephalopathy score[19]).

In round 2 (June and July 2019), this second questionnaire was distributed electronically to the 14 experts from round 1 (online supplemental file 1). A results summary from round 1 was sent to the experts, and the full set of anonymised results were made available at their request. Two email reminders were sent to non-responders. Experts again rated items on a Likert scale and explained their ratings. Responses were analysed as described in round 1, and items meeting ≥80% consensus were kept for the final NeoTree algorithms.

### Patient and public involvement
While key stakeholders were involved in codeveloping the NeoTree digital platform, there was no patient or public involvement in this Delphi study.

### Consent procedures
The goals and processes of the project were explained to the experts in their email invitation, and consent was obtained by email agreement. Experts were verbally informed at the beginning of the first round that their responses would be kept anonymous.

### RESULTS
Twenty-two neonatal experts were invited to participate. Sixteen responded; one declined due to lack of financial incentive, and one declined due to conflict of interest. All respondents had work experience in Africa; one respondent had over 20 years of clinical experience in Malawi and contributed to the development of the COIN guidelines. Demographics of the expert panel are listed in table 1.

### Round 1
Fourteen experts (63% response rate) completed round 1. Interviews averaged 73 min (40–110 min). Thirty-four items (45%) reached consensus (figure 1). These items were either: (1) included, unmodified (32%); (2) included, modified (11%) or (3) changed for clarification in the second round (2%). Items that did not reach consensus (55%) were either: (1) excluded from the revised algorithm (30%); (2) included because they were part of WHO/COIN guidelines or the Thompson score (11%) or (3) changed and submitted for a second round (14%) (table 2).

The expert panel consistently stated that algorithm items must comply with WHO danger signs and COIN guidelines for neonatal sepsis, irrespective of whether the panel agreed with them. For example, experts thought that 'bulging fontanelle' was 'subjective; there are non-infectious causes … many babies' fontanelles bulge

**Table 1** Characteristics of the Delphi panel from round 1

| | Characteristics | n=14 (%) |
|---|---|---|
| Location | Experts from HICs | 9 (64) |
| | Experts from LICs | 5 (36) |
| Level of expertise | Neonatologist | 6 (43) |
| | Paediatrician | 6 (43) |
| | ANNP | 2 (14) |
| Years of neonatal experience following graduate degree (mean±SD) | Overall | 20 (±12) |
| | In LICs | 12 (±7) |
| Work experience in LICs | Africa | 14 (100) |
| | Asia | 7 (50) |
| | Central America | 4 (28) |
| Country of medical degree | UK | 7 |
| | USA | 2 |
| | South Africa | 2 |
| | Rwanda | 1 |
| | Sudan | 1 |
| | Zimbabwe | 1 |

ANNP, advanced neonatal nurse practitioner; HICs, high-income countries; LICs, low-income countries.

when they just cry.' Another item that did not meet consensus was 'poor feeding,' which experts found vague for multiple reasons, including: 'it is too subjective; it depends on how long for … many newborns do not feed well on the first day of life.' However, experts agreed that poor feeding was a sign of possible sepsis if it was 'a new onset of poor feeding when the infant had previously been feeding well.' This item was changed to 'new onset of poor feeding' for the final algorithm.

Two items that were included because they are part of the COIN guidelines highlighted inconsistencies with WHO guidelines. For example, COIN uses a temperature of more than 37.5°C as a fever for a newborn, while WHO and most experts use more than 38°C. Therefore, 37.5°C was included for the Malawian digital platform, but 38°C will be used for other countries. Other items where a difference between the recommendations and the guidelines occurred were antibiotic choice and duration for neonatal sepsis.

Modifications usually involved adopting the language used by WHO or COIN, but there were items that experts felt needed clarifying. For example, experts felt that 'twitching or abnormal movements' needed to be added to the WHO term 'convulsions' because seizures in a neonate can be very subtle. Certain items that could not be revised easily were submitted for the second round according to feedback from the expert panel. For example, experts disagreed that 'very/extremely premature (<32

**Figure 1.** Outcome of algorithm items after round one and round two of Delphi technique.

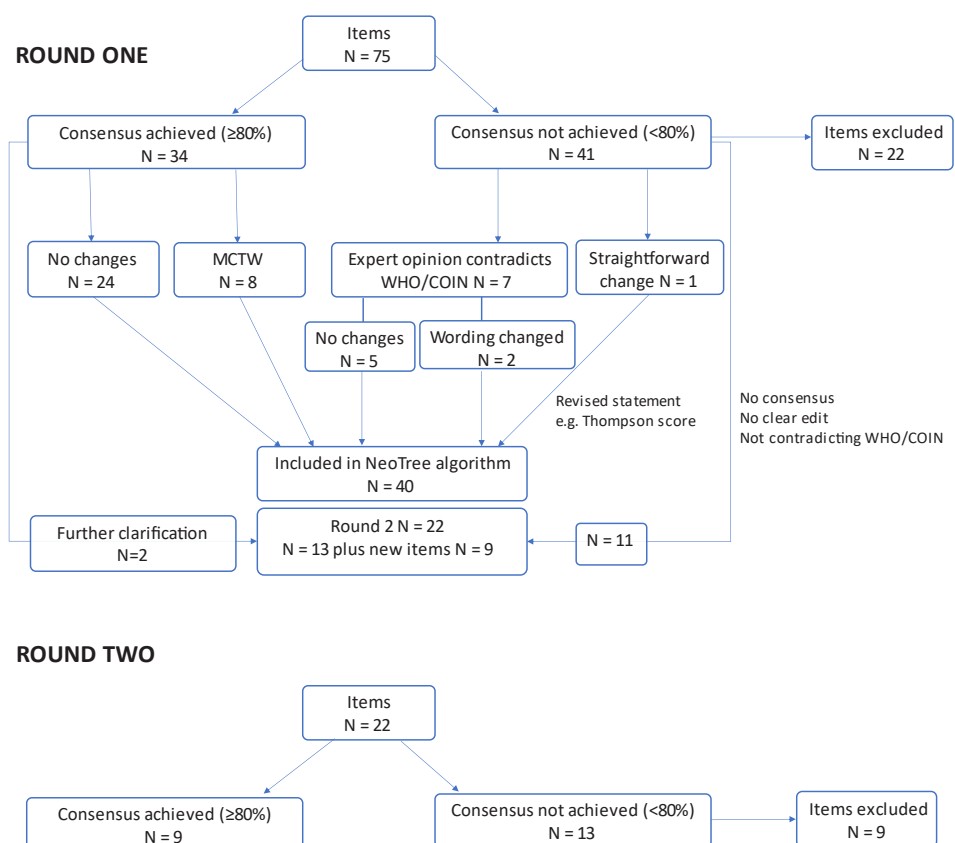

Abbreviations: COIN = Care of the Infant and Newborn in Malawi; MCTW = minor changes to wording; WHO = World Health Organisation

**Figure 1** Outcome of algorithm items after round 1 and round 2 of the Delphi technique. COIN, Care of the Infant and Newborn; MCTW, minor changes to wording.

weeks gestation)' was a major risk factor for sepsis if 'the baby was delivered as a clean cold caesarean section for maternal reasons and the mother was not in labour.' Eighty per cent of experts highlighted that the algorithm should include weight to guide gestation because 'gestation is often unknown' and 'you are relying on [the] Ballard score[20] which has plus or minus 2 weeks accuracy.' Similar opinions regarding method of delivery and the importance of birth weight were expressed for 'slightly premature (32–36 weeks gestation).' Both gestational age brackets were submitted into the second round as risk factors for sepsis after modifying the items to include WHO weight parameters to guide gestation.

Other items that did not gain consensus and were submitted for the second round included items that experts felt needed further clarification. 'Born before arrival' as a minor risk factor for sepsis was clarified to the experts that this meant the baby was born en-route to the hospital (either in a vehicle or on the roadside, both being considered dirty environments in Malawi). A 'neonate admitted with or history of a fever' as a minor risk factor for sepsis was changed to 'mother reports a non-measured fever' in the second round. Lastly, because experts considered the term 'birth injury' unclear, we asked them in round 2 to define what they considered a 'significant birth injury.'

**Table 2** Round 1 heat chart to show which items met consensus and their outcomes

| Subject | Items | Agree, % | Outcome |
|---|---|---|---|
| **Sepsis** | Maternal fever >38°C in labour | 91 | Include |
| **Diagnosis** | PROM >18 hours | 74 | Include (WHO RF) |
| Major RF | Offensive smelling liquor | 74 | Include (WHO RF); MCTW |
| | Very/extremely premature (<32/40 weeks) | 74 | Second round |
| | Prolonged second stage (>3 hours) | 53 | Exclude |
| Minor RF | Prematurity (32–37 weeks gestation) | 81 | Second round |
| | Born before arrival | 70 | Second round |
| Major signs | Boil/abscess | 93 | Include |
| symptoms | Grunting or severe respiratory distress or moderate-severe work of breathing | 97 | Include |
| | Lethargy | 93 | Include |
| | Red skin all around umbilicus | 81 | Include, MCTW |
| | Jaundice <24 hours old | 83 | Include, MCTW |
| | Tachypnoea >60 bpm (>2 hours old) | 83 | Include, MCTW |
| | Convulsions | 89 | Include, MCTW |
| | Pustules all over body | 80 | Include, MCTW |
| | Bulging fontanelle | 77 | Include (WHO danger sign) |
| | Temperature >37.5°C | 73 | Include for Malawi version |
| | Admitted with or history of fever | 68 | Second round |
| | History of apnoea | 67 | Exclude |
| | Bilious vomiting | 61 | Second round |
| Minor signs | Tachypnoea 60–80 bpm and <2 hours old | 85 | Include |
| symptoms | Pallor | 81 | Include |
| | Weak or absent suck (and >34/40 weeks) | 73 | Exclude |
| | Poor feeding | 73 | Include (WHO danger sign), MCTW |
| | Irritability | 70 | Second round |
| | Distended abdomen | 67 | Second round |
| | Heart rate >160 that cannot be explained by fever/crying | 64 | Exclude |
| | Mild work of breathing | 55 | Exclude |
| Additional RF? | Cut-off at 72 hours for early vs late neonatal sepsis? | 100 | Include |
| | Definition of maternal fever >38°C? | 93 | Include |
| | Should PROM be >18 hours or >24 hours in LICs? | 93 | Include>18 hours |
| | Hypothermia <35.5°C | 83 | Second round |
| | Fever in a newborn should be classified as >37.5°C in this setting? | 83 | Include |
| | Please comment on our weighting system of major=100 % / Minor=50% | 77 | Exclude |
| | Cut-off at >34/40 weeks for absent suck as a sign of sepsis? | 60 | Exclude |
| | Reduced movement of limbs | 43 | Exclude |
| | Joint swelling | 42 | Exclude |
| | Criteria for 'consider meningitis' | 42 | Second round |
| Management | Do you agree with the antibiotic doses? | 93 | Include |
| | Do you agree with the specified sepsis investigations if possible? | 83 | Include |
| | Antibiotic duration for symptomatic sepsis=7–10 days? | 83 | Include (change to stop at day seven if clinically well) |
| | Do you agree with the antibiotic choices if no local recommendation? | 80 | Include (add WHO choices) |
| | Antibiotic duration for asymptomatic sepsis=5 days? | 66 | Include for Malawi, exclude for international |
| **Birth** | Resuscitation: BVM >5 mins / CPR>10 mins | 94 | Include |
| **Asphyxia** | Foetal distress | 86 | Include, MCTW |

| Subject | Items | Agree, % | Outcome |
|---|---|---|---|
| **Diagnosis** | Apgar at 5 mins<7 | 78 | Include as per COIN |
| | Vaginal breech | 76 | Exclude |
| | Prolonged second stage >3 hours duration | 73 | Exclude |
| | Vacuum delivery | 69 | Exclude |
| | Emergency caesarean section | 61 | Exclude |
| | Birth injury | 44 | Second round |
| Signs/ symptoms | Convulsions | 93 | Include |
| | Coma | 89 | Include |
| | Lethargy | 84 | Include |
| | Hypotonia and gestation >34/40 weeks | 80 | Include |
| | Irritable | 77 | Exclude |
| | Absent suck and gestation >34/40 weeks | 64 | Second round |
| | Should birth asphyxia be classified as mild, moderate or severe? | 54 | Exclude |
| Additional RF? | Exclude the Moro reflex in LICs due to the difficulties of training HCW in checking safely? | 80 | Include as part of Thompson score |
| | Poor feeding | 49 | Exclude |
| | Respiratory distress | 43 | Include as part of Thompson score |
| | Weight>4 kg | 43 | Exclude |
| Management | Give intravenous fluids if not tolerating oral or nasogastric feeds? | 97 | Include |
| | No passive cooling for infants in LICs? | 94 | Include |
| **RDN Diagnosis** | Do you agree with tachypnoea of >60 bpm for the other categories of RDN? | 100 | Include |
| | Tachypnoea of 60–80 bpm <2 hours old without signs or symptoms of sepsis should be treated as TTN and no antibiotics given? | 76 | Exclude |
| | History of fast/laboured/noisy breathing is relevant as a sign or symptomatic of RDN when not present on admission? | 44 | Exclude |
| RDN diagnostic criteria | Meconium aspiration syndrome | 82 | Include |
| | Transient tachypnoea of the newborn | 66 | Exclude |
| | Respiratory distress syndrome | 63 | Second round |
| | Congenital pneumonia | 63 | Second round |
| Management | Cut-off of 90% $O_2$ saturations before giving oxygen? | 91 | Include |
| | Give antibiotics in all cases except TTN? | 77 | Exclude |
| | Time cut-off of 2 hours for TTN? Would you have a higher or lower threshold? | 54 | Exclude |
| **Hypothermia** | Diagnostic criteria for hypothermia? | 100 | Include |

BPM, breaths per minute; BVM, bag valve mask; COIN, Care of the Infant and Newborn; CPR, cardiopulmonary resuscitation; HCW, heathcare worker; LICs, low-income countries; MCTW, minor changes to wording; PROM, prolonged rupture of membranes; RDN, respiratory distress of the newborn; RF, risk factor; TTN, transient tachypnoea of the newborn.

## Key findings by algorithm

The first important finding was that the 'major' or 'minor' algorithmic weighting system (where one major risk factor for sepsis is equivalent to two minors) used to diagnose neonatal sepsis was near consensus (77%) but did not meet the 80% threshold. Experts called for further evidence before adopting this system: 'It is a difficult thing to do…you need to work out how specific and sensitive the app is by looking at blood cultures.' Two experts suggested using a sepsis risk score calculator, and another two experts highlighted that WHO only uses danger signs. This weighting system was subsequently removed from the algorithm.

The second significant algorithmic finding was on HIE. An academic expert in neonatal encephalopathy discouraged the use of the term 'birth asphyxia,' a term used by Malawian HCWs and therefore incorporated into the original algorithm.

You really must not call it birth asphyxia because birth asphyxia means failing to breathe at birth and what you are talking about is encephalopathy.

Additional feedback on the algorithm focused on the combination of risk factors or clinical signs and symptoms to consider or diagnose HIE. Experts cited a lack of evidence for using risk factors to diagnose birth asphyxia

and that the digital platform should only be using clinical signs and symptoms.

> Birth asphyxia is not about risk factors. If you have encephalopathy, it is a clinical diagnosis, and it is irrelevant what your risk factors are.

Experts recommended using a validated encephalopathy score,[19 21] which was incorporated into the HIE algorithm. The risk factors that met consensus may be used as prompts to perform the Thompson score, which uses clinical signs and symptoms exclusively to diagnose HIE.

Third, for the respiratory algorithm, experts highlighted that 'It is hard to make an accurate diagnosis of a respiratory condition without investigations.' Therefore, the algorithm should focus instead on the management of respiratory distress. All respiratory conditions (respiratory distress syndrome, meconium aspiration, congenital pneumonia and transient tachypnoea of the newborn (TTN)) now fall under the umbrella diagnosis of respiratory distress of the newborn within the algorithm. For teaching purposes, the four respiratory conditions will be included as 'diagnoses to consider' in the management.

Finally, for the hypothermia algorithm, experts commented that first-line treatment for all newborns be skin-to-skin care including those who were severely hypothermic (<32°C) unless they showed any signs or symptoms of being unstable. Additionally, experts did not think it was realistic to review a newborn every 15–30 min when hypothermic. No major revisions were made to the hypothermia algorithm.

## Round 2

Ten (71%) experts completed round 2, seven electronically and three by telephone interview. Four experts dropped out (three from HICs, one from LIC); three did not respond to email reminders and one expert was unable to meet the completion deadline. Nine items (41%) reached consensus (figure 1). These items were either (1) included, unmodified (36%) or (2) included, modified in the revised algorithm (5%). Items that did not reach consensus (59%) were either (1) excluded

**Table 3** Round 2 heat chart to show which items met consensus and their outcomes

| Subject | Items | Agree, % | Outcome |
|---|---|---|---|
| **Sepsis diagnosis** | <32/40 weeks gestation and/or<1500 g | 82 | Include |
| | 32–36/40 weeks gestation and/or 1500–2500 g | 62 | Exclude |
| 'Other' RF | Babies born en route to the hospital | 54 | Exclude |
| | Mother reports a non-measured fever | 52 | Exclude |
| Major sign | Bilious vomiting with severe abdominal distension | 88 | Include |
| 'Other' sign/ | Irritable/inconsolable baby | 64 | Exclude |
| symptoms | Do you think a one-off T<35.5°C should be added as a sign of neonatal sepsis in an LIC? | 44 | Exclude |
| Additional RF? | Swollen red eyelids with pus | 88 | Include |
| | Unconscious | 86 | Include, MCTW |
| | Central cyanosis | 78 | Include (WHO danger sign) |
| | Poor capillary refill or perfusion | 68 | Exclude |
| | Does the baby look ill? | 60 | Exclude |
| **Consider** | Drowsy, lethargic or unconscious with T>37.5°C | 96 | Include |
| **Meningitis** | Bulging fontanelle with T>37.5°C | 94 | Include |
| | Irritability with a high-pitched cry with T>37.5°C | 92 | Include |
| | Abnormal movements/twitching or convulsions with T>37.5°C | 90 | Include |
| | Abnormal tone with T>37.5°C | 80 | Include |
| **HIE diagnosis** | Do you agree with absent suck and gestation <32/40 weeks as a sign of HIE? | 42 | Exclude |
| | How should we describe 'significant' birth injury as a risk factor for HIE? | 20 | Exclude |
| **RDN diagnosis** | Do you agree with gestation <34/40 for part of the diagnostic criteria for respiratory distress syndrome? | 74 | Include, change to WHO definition |
| | Do you agree with coarse crackles (instead of unilateral crackles) for part of the diagnostic criteria for congenital pneumonia? | 74 | Include, change to WHO wording |
| | Do you agree with T>37.5°C or <36.5°C for part of the diagnostic criteria for congenital pneumonia? | 70 | Include, change to expert suggestion |

HIE, hypoxic ischaemic encephalopathy; LICs, low-income countries; MCTW, minor changes to wording; RDN, respiratory distress of the newborn; RF, risk factor; T, temperature.

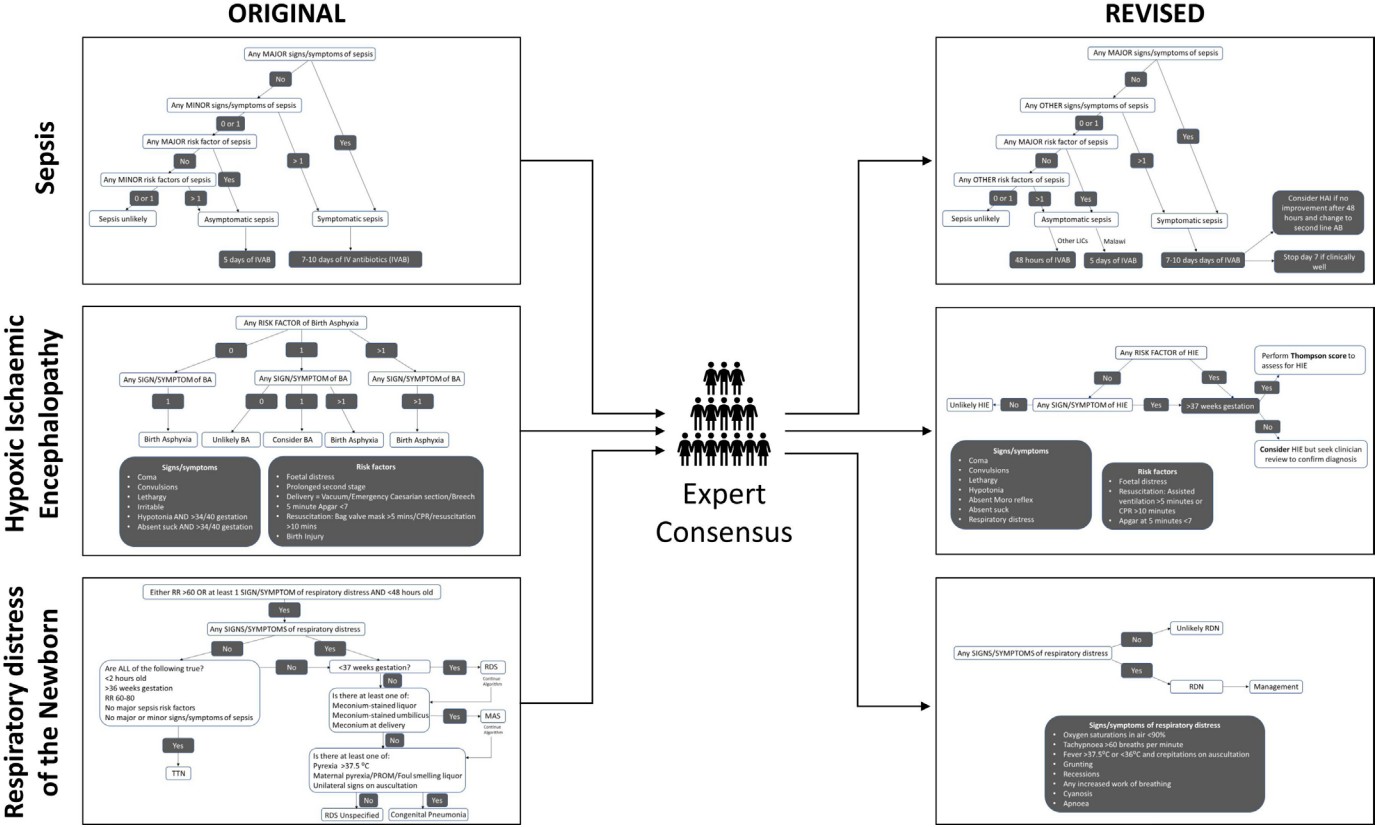

**Figure 2** Modification of the algorithms as a result of the Delphi technique. AB, antibiotics; BA, birth asphyxia; CPR, cardiopulmonary resuscitation; HAI, hospital-acquired infection; HIE, hypoxic ischaemic encephalopathy; LICs, low-income countries; MAS, meconium aspiration; RDN, respiratory distress of newborn; RDS, respiratory distress syndrome; RR, respiratory rate; TTN, transient tachypnoea of newborn.

(41%) or (2) included, modified according to WHO guidelines or expert suggestion (18%) (table 3).

In round 1, experts indicated that hypothermia was a major sign of sepsis and should be included in the sepsis algorithm if persistent. In round 2, we clarified that the digital platform is to be used at the time of admission onto the neonatal unit, at which point the HCW will only have one temperature reading. Experts in round 2 disagreed that a single temperature reading of <35.5°C was a sign of sepsis and felt that it would more likely be due to environmental hypothermia, a common problem in LICs. Additionally, in round 2, it was established that experts were much more concerned with extremely premature and/or<1500 g neonates (88% consensus) compared with slightly premature and/or 1500–2500 g neonates (62% consensus) being at risk for neonatal sepsis. Central cyanosis was an addition to the second round as an expert suggestion to include all WHO danger signs; despite missing consensus (with 78% agreement), it was ultimately included in the final sepsis algorithm to comply with WHO guidelines for danger signs.

All of the respiratory items marginally missed consensus. All three items were still included in the education sections of the digital platform with revisions in line with WHO diagnostic criteria. Modifications to the algorithms can be found in figure 2 and online supplemental file 2.

## DISCUSSION

We report the use of a modified Delphi technique to review digital clinical pathway algorithms for four neonatal conditions managed by HCWs in LICs. Approximately two-thirds (62%) of the original algorithm items were ultimately included for use in the NeoTree digital platform based on consensus expert opinion and national/international guidelines. The NeoTree team revised the algorithms based on this feedback. Expert discussion emphasised gaps in evidence in neonatal care in LICs, highlighting areas for future research.

Each algorithm had components that triggered debate among the experts. For neonatal sepsis, three points were discussed. First, experts called for further evidence before adopting a 'major' and 'minor' algorithmic weighting system to diagnose neonatal sepsis. In response, the NeoTree research team are conducting a study in Zimbabwe and Malawi looking at which clinical indicators are predictors of positive blood cultures. Second, there was disparity in opinion regarding whether to give prophylactic antibiotics and the duration of antibiotics for newborns with risk factors for sepsis who remain clinically well without any supporting investigations (NeoTree's equivalence to asymptomatic sepsis). The WHO recommendation to administer prophylactic antibiotics for a neonate with maternal risk factors for sepsis is considered

weak with very low-quality evidence.[22] Despite reaching a consensus on particular risk factors (prolonged rupture of membranes, maternal fever), experts also highlighted the evidence base as weak. In terms of duration of treatment for asymptomatic sepsis, while expert opinion varied, the Malawian guidelines recommend a 5-day course[11] while WHO recommends 2 days.[15] The NeoTree algorithms will therefore keep to local and international recommendations, but the NeoTree team will feed back to the Malawian COIN expert panel that consensus suggested 5 days is too long to treat newborns with sepsis risk factors only. Third, experts disputed the treatment choice and duration for symptomatic neonatal sepsis; incidentally, WHO recommendations lack strong evidence or efficacy.[17]

For the HIE algorithm, the Thompson score was preferred because it is simpler to perform, less time consuming and better at predicting poor outcomes in moderate and severe HIE during the first hours of life compared with the Sarnat score at 24 hours.[19] The NeoTree research team suspected that measures such as examining for posturing and Moro reflex would be relatively complicated for frontline HCWs with minimal training to assess. However, neonatal experts' experience and previous studies in LICs[23] assured the team that the score is relatively straightforward to teach.

Several points of discussion also centred on the respiratory algorithms. First, experts noted that even with investigations in HICs respiratory conditions may be difficult to diagnose.[24] Second, despite experts' concerns about antibiotic overprescribing in LICs and the need to differentiate TTN from other respiratory conditions, they did not think this was currently feasible in LICs due to limitations in HCW capacity, resources and knowledge. Thus, experts agreed that all neonates with signs of respiratory distress should have respiratory support and antibiotics. A recent study justified the use of antibiotics for tachypnoea alone in a neonate in a resource-limited setting.[25] Third, experts recommended performing chest X-rays (if available) only if imaging would change management (eg, a longer course of antibiotics for congenital pneumonia) or if the neonate was deteriorating.

With the proliferation of clinical digital platforms in HICs and LICs, there is growing concern with the quality and safety standards of their clinical guidance. Countries and organisations (including WHO) are now taking measures to ensure application developers fulfil a strict set of criteria to protect patients.[26] While the Delphi technique can establish expert consensus, it may also strengthen the safety and quality standards of clinical algorithms. This technique has been widely used in developing paper-based neonatal clinical guidelines in HICs and LICs.[27–29] There are also studies that have used the Delphi technique to develop items used in m-health tools.[30–32] Our study is unique in the application of this technique to develop algorithms on a digital platform specific to neonatal care in low-resource settings.

This study has several limitations. The choice of using a modified two-step Delphi process meant that a final face-to-face meeting was not possible, which may have prevented some exchange of important information to clarify differences in expert opinion. However, this method allowed for the contributions of geographically dispersed experts, maintained their anonymity and prevented them from conforming to other experts. The recruitment of more experts from HICs (64%) compared with LICs (36%), despite originally inviting equal numbers to participate, could have contributed to expert panel bias. We invited three Malawian clinicians as experts; the absence of their input could be another limitation, since the algorithms had initially been contextualised to the Malawian setting. However, the end goal of NeoTree development is to be applicable in a wide range of resource-limited settings; therefore, experts with a broad geographical range of clinical experience were recruited. Drop-outs from the first to the second round could have affected the consensus level and contributed to attrition bias.

Some factors may have contributed to selection bias. The Delphi process is time intensive, which could have meant that those clinicians who are busier with perhaps even more clinical expertise or those with limited internet access (mainly LICs) could not participate. Additionally, offering a financial incentive might have obtained a more equal representation of experts. Another drawback of the Delphi being a labour-intensive process was that a year elapsed between the two rounds. Experts may have forgotten the algorithms and items from the first round in the second round if they did not read the summary of results or refresh their knowledge of the algorithms. Experts reported that they found the layout of the second questionnaire confusing; a redesign contributed to delays.

This study used the Delphi technique to refine four clinical decision algorithms in a neonatal digital platform designed for HCWs in LICs to standardise and improve the quality of newborn care. The key to implementing the NeoTree algorithms in other LICs will be to demonstrate that clinical algorithms in a digital platform versus paper-based guidelines can aid HCWs in making faster, more accurate diagnoses and provide better, more cost-effective treatment that will ultimately improve the quality of newborn care and reduce mortality. This will require a large-scale clinical-trial evaluation. Ultimately, with consensus opinion shaping the algorithms of this digital platform, accurate data capture, immediate clinical assessment and optimal medical care may be achieved to improve neonatal outcomes.

**Acknowledgements** We thank all the participants of the Delphi surveys for their invaluable contributions to revise the clinical algorithms and highlight open questions, including Dr Helen Brotherton, Dr Simbarashe Chimhuya, Dr Cath Harrison, Dr Tyler Hartmann, Dr Hammad Khan, Dr Camilla Kindgon, Dr Tom Lissauer, Dr Azza Mashumba, Kathy Mellor (MBE), Professor Elizabeth Molyneux, Dr Cally Tann, Dr Cliff O'Callahan, Susan Quinton, and Dr Raissa Teteli. We would also like to thank Dr Brian Corden and Dr Alice Myers for their reviews of the manuscript.

**Contributors** CC led the development of the original algorithms refined in this paper, supervised by MH.ME, CC and MH conceived the study, generated the

methodology and designed the questionnaires. ME conducted the interviews and analysed the data. ME and MHC produced the first draft and contributed equally to this manuscript. CC, FF and MH provided edits and comments to the draft. All authors reviewed and approved the final version of the manuscript.

**Funding** This work was supported in part by the Wellcome Trust Digital Innovation Award (215742/z19/z). FF is supported by the Academy of Medical Sciences, the funders of the Starter Grant for Clinical Lecturers scheme and the UCL Great Ormond Street NIHR Biomedical Research Centre.

**Disclaimer** The funders had no role in the study design, data collection and analysis, or preparation of the manuscript.

**Competing interests** CC and MH are cofounders of the NeoTree platform and continue to conduct research related to its development. FF and MH are trustees of the NeoTree Foundation. The NeoTree platform is a not-for-profit product; none of the coauthors benefit financially.

**Patient consent for publication** Not required.

**Ethics approval** Ethics approval was not required for this study according to the University College of London Research Ethics Committee.

**Provenance and peer review** Not commissioned; externally peer reviewed.

**Data availability statement** Data are available on reasonable request. All data relevant to the study are included in the article or uploaded as online supplemental information.

**ORCID iDs**
Mari Evans http://orcid.org/0000-0001-5113-3629
Mark H Corden http://orcid.org/0000-0001-6146-1634
Caroline Crehan http://orcid.org/0000-0002-3655-6954
Felicity Fitzgerald http://orcid.org/0000-0001-9594-3228
Michelle Heys http://orcid.org/0000-0002-1458-505X

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
