## [Reviewer comments · BMJ Open]

ARTICLE DETAILS

TITLE (PROVISIONAL)	Refining clinical algorithms for a neonatal digital platform for low-income countries: A modified Delphi technique
AUTHORS	Evans, Mari; Corden, Mark; Crehan, Caroline; Fitzgerald, Felicity; Heys, Michelle

VERSION 1 – REVIEW

REVIEWER	Pelle, Karell Foundation for Innovative New Diagnostics
REVIEW RETURNED	19-Aug-2020

GENERAL COMMENTS	Congratulations to the authors on putting together this manuscript for four neonatal care algorithms for LICs, starting with Malawi. The article highlights the need for more evidence for neonatal guidelines in LICs and the complexity of synthesizing available evidence into useful clinical decision algorithms. Minor revisions/comments: • I would request the authors expand the Methods sections to elaborate how the Likert scale was used to assess agreement vs disagreement.• Could the authors expand on the final algorithms? As stated, 38% of the algorithm original items were rejected through the Delphi process. From a pragmatic point of view at lower levels of the health care system in LICs, busy health facilities, it is probably more feasible to collect fewer data points to reach a clinical classification. However, has the algorithm suffered in its clinical efficacy? and would the use of a digital platform help reach a sweet spot?• I appreciate the discussions on the limitations of the study. An additional limitation is the lack of representation of Malawian clinicians in the expert group, especially since the algorithms are contextualized to the Malawian context. Could the authors expand on this?• Figure 2: it is quite informative for readers to see the original and the consensus-based algorithms for sepsis side by side. The authors should do the same for the 3 other algorithms.
--

REVIEWER	Dufendach, Kevin Cincinnati Children's, United States
REVIEW RETURNED	22-Jan-2021

GENERAL COMMENTS	In this article, the researchers describe their process of refining neonatal clinical algorithms using two rounds of a modified Delphi technique. The researchers interviewed 14 neonatal clinicians to determine items that should be included in the final clinical decision
--

support algorithm. Overall, the process was described reasonably well, and the results are important in that they present an evaluation of current algorithms by expert opinion in a relatively rigorous manner. One weakness of this approach is that the strength of recommendations is not based directly on clinical trials or population-level patient data (e.g. as used in the Sepsis Calculator from Kaiser Permanente <https://neonatalesepsiscalculator.kaiserpermanente.org/>), although presumably the expert clinicians are basing their recommendations on scientific evidence.

While I found the research process and results interesting, I do think this article could use some major revisions to improve its potential impact. To me, the most glaring omission from the article is the clear presentation of the results from the second round of feedback. Figure 2 presents the modifications of the Sepsis algorithm, but the remaining decision trees are relegated to the online content. In contrast, the initial round one table is given a full-color multi-page table. The impact of this research could be significantly improved by making the final results easier to access.

The following comments were more minor points and comments from the article:

1. The title identifies a “neonatal digital platform,” which is identified as NeoTree, but the focus of the research is on the refinements of the algorithms. Online material demonstrate screenshots of NeoTree, but this is really secondary to the research. It would be appropriate to provide the scope in the introduction (i.e. focus on the constraints and desires of the final delivery mechanism) and then comment on the implementation plan, using NeoTree.
2. Please use the correct “≥” symbol. This is not a “greater-than” with underlining. In Word, this can be found in Insert  Symbol
3. In Abstract □ Results section, the sentence “Experts consistently stated that items must be in line with local and WHO guidelines (irrespective of the level of supporting evidence or expert opinion).” Is unclear without the context of the remainder of the article. Please consider a revision of this sentence.
4. The statistic “3/4 of the world now own mobile phones” could be further strengthened by calling out the use in the target population. In this case, the target population is health care providers in LIC’s. Gathering background data on that population would help to strengthen this statement, as ¾ seems low, at least for adults.
5. Page 9, line 40. Please clarify agreement for inclusion □ “(≥80% agreement for inclusion)”
6. P. 11, line 44. The Thompson score is identified for the first time in results. This should be mentioned in the methods section.
7. P. 12, line 47. The Ballard score is mentioned in a comment without a corresponding citation. Please add citation.
8. Table 2 (round one items table/heatmap):
 - a. Consider using a full gradient for the heat map colors as opposed to blocks of percentile agreement.
 - b. Consider how a heat map will appear when the article is printed or copied in black and white. Some results may not appear correctly. The black text on red background may not provide adequate contrast to read, even with the current colored version.
 - c. P. 14 line 9, consider adding the descriptor “increased” to read “Mildly increased work of breathing.” However, this might not be reasonable if this was the wording presented to the clinical experts.
9. P. 16, line 12: Comma needed □ “If you have encephalopathy, it is a clinical...”

	10. P. 16, line 26: “It is hard to make” should begin with a capital “I” since it is starting a sentence in the quotation. 11. P. 17, line 22: Add comma □ “In round two, we clarified...” Also, p. 17 line 31. 12. P. 17, lines 33-35: <1500g is a “very low birthweight infant” as opposed to “extremely premature” infant. 1500-2500g is a “low birthweight infant” as opposed to “slightly premature.” I do recognize that this article uses birthweight as a surrogate for gestational age, but that should then be clarified during this part. 13. P 18, line 22: Add comma □ “In response, the NeoTree...” 14. P. 18, line 36: The word “only” is unclear as to what it is modifying. 15. In acknowledgements, the “participants” are thanked. With 14 clinicians, is there a reason their names could not be shared (with their permission)? Since this research represents expert consensus, it seems relevant to know who the experts were. As these results directly influence the design of a product, an acknowledgement of the authors’ affiliation and role in the NeoTree application would be appropriate in the “competing interests” section. Thank you for the opportunity to review this manuscript. I would welcome the opportunity to review any subsequent revisions of this work.
--	---

VERSION 1 – AUTHOR RESPONSE

Manuscript Section	Reviewer comment	Response
Overall	One weakness of this approach is that the strength of recommendations is not based directly on clinical trials or population-level patient data (e.g. as used in the Sepsis Calculator from Kaiser Permanente https://neonatalespsiscalculator.kaiserpermanente.org/), although presumably the expert clinicians are basing their recommendations on scientific.	The purpose of the Delphi technique is to utilize expert opinion where “clinical trials or population-level patient data” do not exist. The nature of the setting in which our neonatal digital platform is being deployed precludes the employment of known clinical guides for the management of neonates.
	Please use the correct “≥” symbol. This is not a “greater-than” with underlining. In Word, this can be found in Insert  Symbol	Thank you for this comment. The manuscript has been edited accordingly.
Abstract	In Abstract ◊ Results section, the sentence “Experts consistently stated that items must be in line with local and WHO guidelines (irrespective of the level of supporting evidence or expert opinion).” Is unclear without the context of	Thank you for this comment. The manuscript has been edited accordingly.

	the remainder of the article. Please consider a revision of this sentence.	
Introduction	The title identifies a “neonatal digital platform,” which is identified as NeoTree, but the focus of the research is on the refinements of the algorithms. Online material demonstrate screenshots of NeoTree, but this is really secondary to the research. It would be appropriate to provide the scope in the introduction (i.e. focus on the constraints and desires of the final delivery mechanism) and then comment on the implementation plan, using NeoTree.	Thank you for this comment. The manuscript has been edited accordingly.
	The statistic “3/4 of the world now own mobile phones” could be further strengthened by calling out the use in the target population. In this case, the target population is health care providers in LIC’s. Gathering background data on that population would help to strengthen this statement, as ¾ seems low, at least for adults.	This sentence has now been removed after addressing the restructuring of the introduction, per the reviewer’s suggestion.
Methods	I would request the authors expand the Methods sections to elaborate how the Likert scale was used to assess agreement vs disagreement.	Thank you for this comment. The manuscript has been edited accordingly.
	Page 9, line 40. Please clarify agreement for inclusion ◊ “(≥80% agreement for inclusion)”	Thank you for this comment. The manuscript has been edited accordingly.
Results	Could the authors expand on the final algorithms?	In response to feedback from the other reviewer, we have highlighted changes to the algorithms by moving some of the online supplementary material to Figure 2.
	As stated, 38% of the algorithm original items were rejected through the Delphi process. From a pragmatic point of view at lower levels of the health care system in LICs, busy health facilities, it is probably more feasible to collect fewer data points to reach a clinical classification. However, has the algorithm suffered in its clinical efficacy?	Thank you for these important points. The purpose of NeoTree was to be more sensitive and specific than basic neonatal guidelines used in LICs and thus required more clinical data points. The clinical efficacy of the platform has not yet been evaluated and will be the subject of future study.
	and would the use of a digital platform help reach a sweet spot?	Indeed, we intend to evaluate this point as well with future studies.
	To me, the most glaring omission from the article is the clear presentation of the results from the	Thank you for this comment. We have edited

	second round of feedback. Figure 2 presents the modifications of the Sepsis algorithm, but the remaining decision trees are relegated to the online content. In contrast, the initial round one table is given a full-color multi-page table. The impact of this research could be significantly improved by making the final results easier to access.	Figure 2 to include the changes between the initial and the final algorithms for three of the four diagnoses that were reviewed. We have included the outcomes of the second round in a new Table 3.
	P. 11, line 44. The Thompson score is identified for the first time in results. This should be mentioned in the methods section.	Thank you for this comment. The manuscript has been edited accordingly.
	P. 12, line 47. The Ballard score is mentioned in a comment without a corresponding citation. Please add citation.	Thank you for this comment. The manuscript has been edited accordingly.
	Table 2 (round one items table/heatmap): a. Consider using a full gradient for the heat map colors as opposed to blocks of percentile agreement.	Thank you for this comment. A full colour gradient for the heat map has been added.
	Table 2 (round one items table/heatmap): b. Consider how a heat map will appear when the article is printed or copied in black and white. Some results may not appear correctly. The black text on red background may not provide adequate contrast to read, even with the current colored version.	Thank you for this comment. We have test printed in black and white to ensure the contrast is adequate.
	P. 14 line 9, consider adding the descriptor “increased” to read “Mildly increased work of breathing.” However, this might not be reasonable if this was the wording presented to the clinical experts.	Thank you for this comment. Indeed, this was the wording presented to clinical experts. Therefore, we have not edited the manuscript. We suspect that there may be cross-cultural differences between how the phrase “work of breathing” is utilized, as in the UK any presence of work of breathing suggests a certain level of respiratory distress, while in the US work of breathing is only significant if increased.
	P. 16, line 12: Comma needed ◊“If you have encephalopathy, it is a clinical...”	Thank you for this comment. The manuscript has been edited accordingly.

	P. 16, line 26: "It is hard to make" should begin with a capital "I" since it is starting a sentence in the quotation.	Thank you for this comment. The manuscript has been edited accordingly.
	P. 17, line 22: Add comma ◊ "In round two, we clarified..." Also, p. 17 line 31.	Thank you for this comment. The manuscript has been edited accordingly.
	P. 17, lines 33-35: <1500g is a "very low birthweight infant" as opposed to "extremely premature" infant. 1500-2500g is a "low birthweight infant" as opposed to "slightly premature." I do recognize that this article uses birthweight as a surrogate for gestational age, but that should then be clarified during this part.	Thank you for this comment. The manuscript has been edited accordingly.
	P 18, line 22: Add comma ◊ "In response, the NeoTree..."	Thank you for this comment. We have edited the manuscript accordingly.
	P. 18, line 36: The word "only" is unclear as to what it is modifying.	Thank you for this comment. We have edited the manuscript accordingly.
Discussion	I appreciate the discussions on the limitations of the study. An additional limitation is the lack of representation of Malawian clinicians in the expert group, especially since the algorithms are contextualized to the Malawian context. Could the authors expand on this?	Thank you for this comment. The manuscript has been edited accordingly in two locations – the demographics of the experts in Results, and the Discussion.
	Figure 2: it is quite informative for readers to see the original and the consensus-based algorithms for sepsis side by side. The authors should do the same for the 3 other algorithms.	Thank you for this comment, which was echoed by the other reviewer. We have edited the Figure accordingly. No major changes were made to the Hypothermia algorithm from this study so this algorithm remains in the online supplementary material. This is also noted in the manuscript.
Acknowledgements	In acknowledgements, the "participants" are thanked. With 14 clinicians, is there a reason their names could not be shared (with their permission)? Since this research represents expert consensus, it seems relevant to know who the experts were.	Thank you for this comment. We have edited the manuscript accordingly.

	As these results directly influence the design of a product, an acknowledgement of the authors' affiliation and role in the NeoTree application would be appropriate in the "competing interests" section.	Thank you for this comment. We have edited the manuscript accordingly.
--	--	--

VERSION 2 – REVIEW

REVIEWER	Dufendach, Kevin
REVIEW RETURNED	19-Mar-2021

GENERAL COMMENTS	Thank you for the opportunity to review this revision. I appreciate the responses and revisions from the authors, and I think they have significantly improved the manuscript. At this point, the article reads much more cohesively. The emphasis is now much more on the final outcomes and revisions of the model, although the authors also make it very clear what content is being made available in the NeoTree App. I have some relatively minor edits and recommendations: P.3, line 30: Add "for inclusion" for clarity  items meeting consensus for inclusion ... (or could consider a synonym so as not to repeat "inclusion/included" right next to each other P3, line 49-50: This still reads awkwardly. Consider, "Per experts' recommendations, items in line with local and WHO guidelines yet not meeting consensus were still included to encourage consistency for frontline healthcare workers." P6, line 10: remove "been" unless you can identify who or what has contributed to the decreased mortality rate  mortality rate has halved since 1990 P6, line 11: add hyphen for "two-thirds" p.6, line 33: Add comma after "countries"  from high-income countries, as there is often... p.9, line 14: Need comparator for "less confusing." It's less confusing than what? A 10-point scale? A 4-point scale? p.9, line 31: There's a leftover underline under the \geq sign. Also elsewhere in the manuscript. p.10, line 47: How were the years counted for the ANNP's, who did not complete medical school? p.12, line 17: the word "that" is repeated
---

VERSION 2 – AUTHOR RESPONSE

bmjopen-2020-042124.R1 - "Refining clinical algorithms for a neonatal digital platform for low-income countries: A modified Delphi technique"

Reviewer comment	Author response
P.3, line 30: Add "for inclusion" for clarity  items meeting consensus for inclusion ... (or could consider a synonym so as not to repeat "inclusion/included" right next to each other	Thank you for this comment. The manuscript has been edited accordingly.
P3, line 49-50: This still reads awkwardly. Consider, "Per experts' recommendations, items in line with local and WHO guidelines yet not meeting consensus were still included to encourage consistency for frontline healthcare workers."	Thank you for this comment. The manuscript has been edited accordingly.

P6, line 10: remove “been” unless you can identify who or what has contributed to the decreased mortality rate  mortality rate has halved since 1990	Thank you for this comment. The manuscript has been edited accordingly.
P6, line 11: add hyphen for “two-thirds”	Thank you for this comment. The manuscript has been edited accordingly.
p.6, line 33: Add comma after “countries”  from high-income countries, as there is often...	Thank you for this comment. The manuscript has been edited accordingly.
p.9, line 14: Need comparator for “less confusing.” It’s less confusing than what? A 10-point scale? A 4-point scale?	Thank you for this comment. The manuscript has been edited accordingly.
p.9, line 31: There’s a leftover underline under the \geq sign. Also elsewhere in the manuscript.	Thank you for this comment. The manuscript has been edited accordingly.
p.10, line 47: How were the years counted for the ANNP’s, who did not complete medical school?	Thank you for this comment. The manuscript has been edited accordingly.
p.12, line 17: the word “that” is repeated	Thank you for this comment. The manuscript has been edited accordingly.